# Gamma-ray Irradiation of Rodent Diets Alters the Urinary Metabolome in Rats with Chemically Induced Mammary Cancer

**DOI:** 10.3390/metabo12100976

**Published:** 2022-10-16

**Authors:** Jeevan K. Prasain, Landon S. Wilson, Clinton Grubbs, Stephen Barnes

**Affiliations:** 1Department of Pharmacology and Toxicology, University of Alabama at Birmingham, Birmingham, AL 35294, USA; 2Targeted Metabolomics and Proteomics Laboratory, University of Alabama at Birmingham, Birmingham, AL 35294, USA; 3Department of Surgery, University of Alabama at Birmingham, Birmingham, AL 35294, USA

**Keywords:** irradiated diet, metabolomics, chemoprevention, MNU, breast cancer

## Abstract

In this study, a comparative, untargeted metabolomics approach was applied to compare urinary metabolite profiles of rats fed irradiated and non-irradiated diets. γ-Irradiated and non-irradiated NIH 7001 diet was given orally to animals beginning 5 days after exposure to the carcinogen N-methyl-N-nitrosourea and continued for 120 days. There was a 36% reduction in mammary tumor incidence in rats consuming the γ-irradiated diet, compared to rats receiving the non-irradiated form of the same diet. Urine samples from rats fed with γ-irradiated and non-irradiated diets were analyzed using nanoLC-MS/MS on a Q-TOF mass spectrometer, collecting positive and negative ion data. Data processing involved feature detection and alignment with MS-DIAL, normalization, mean-centering and Pareto scaling, and univariate and multivariate statistical analysis using MetaboAnalyst, and pathway analysis with Mummichog. Unsupervised Principal Component Analysis and supervised Partial Least Squares-Discriminant Analysis of both negative and positive ions revealed separation of the two groups. The top 25 metabolites from variable importance in projection scores >1 showed their contributions in discriminating urines the γ-irradiated diet fed group from non-irradiated control diet group. Consumption of the γ-irradiated diet led to alteration of several gut microbial metabolites such as phenylacetylglycine, indoxyl sulfate, kynurenic acid, hippurate and betaine in the urine. This study provides insights into metabolic changes in rat urine in response to a γ-irradiated diet which may be associated with mammary cancer prevention.

## 1. Introduction

Food irradiation to eliminate/reduce microorganisms has become standard practice in many countries worldwide [1,2]. The potential of food irradiation to prevent infectious diseases and to extend the shelf life of perishable foods by reducing or eliminating microorganisms has been recognized by the CDC [3] and WHO [4]; however, not much is known about potential for long-term health consequences after consuming irradiated diets [5]. Since many countries permit irradiation of foods and their consumption is widespread, it is important to address consumers concern as to whether consumption of such diets has an impact on chronic diseases such as cancer [6]. Furthermore, diet composition is associated with changes in the gut microbiota and their diversity. The latter is considered to be important in the risk, prevention or treatment of cancer. Breast cancer is the second most common cancer world wide; therefore, development of effective preventative strategies is of critical importance. There is a large body of evidence linking diet and nutrition with the development of mammary tumors [7,8,9]. This is an area where intervention and education can have a major preventive effect on the occurrence of cancer on a worldwide basis. While diet-based influences in cancer chemoprevention are recognized [10], the compounds coming from the diet or produced secondarily by the microbiota are not well understood. Both nutritive or non-nutritive components (e.g., phytochemicals) present in the diet can influence the process of carcinogenesis [11,12].

Secondarily, commercial animal diet manufacturers in the past decade have introduced γ-irradiation of diets with ^60^Co to eliminate disease-causing microbial contamination and many institutional animal resource programs have changed to using these γ-irradiated diets (gIDs). Besides eliminating microbial contamination, γ-irradiation also causes chemical modifications of food components [13] and therefore, doubts about safety or beneficial effects of γ-irradiated foods continue to be debated in public as well as in scientific circles [5]. Although Su et al. have reported the effects of autoclaved or irradiated food on mouse urine and fecal metabolomics [14], the current knowledge on the interplay between consumption of sterilized diets by radiation and cancer biology is limited.

We have previously shown that ^60^Co γ-irradiation causes substantial changes in the animal diet metabolome, particularly the oxidation of medium chain fatty acids and linoleic acid-derived oxidation products [13]. While there has been growing interest in the food metabolome [15], there have been no reports examining differences in the physiologic metabolomes between animals chronically fed with γ-irradiated and non-irradiated diets. Metabolomics- a comprehensive analysis of all metabolites in a biological sample may provide the link between metabolic alteration and chemoprevention. Metabolic shifts resulting from alteration of diet may be associated with carcinogenesis or cancer prevention [16]. In the present study, we sought to understand whether consumption of a gID affects the urinary metabolite profile, which by implication may be associated with increased cancer risk or prevention.

## 2. Experimental Design

Animal experiments were conducted in facilities at the University of Alabama at Birmingham (Birmingham, AL, USA). All experimental procedures were conducted under the oversight and approval of the UAB Institutional Animal Care and Use Committee and in accordance with the Guide for the Care and Use of Laboratory Animals (IACUC # 20269). All animals were housed 5/cage in a room lighted 12 h/day and maintained at 22 °C. Chemoprevention studies were performed as previously described with some modifications [17] in female Sprague Dawley rats. A batch of NIH 7001 diet was divided into two parts by the manufacturer Envigo (Indianapolis, IN, USA); one was treated with 20–50 Gy of ^60^Co γ-irradiation (gID) and the other, non-irradiated (non-gID), was used as the control diet. At day 50, all animals were treated with a single dose of N-methylnitrosourea (MNU) (50 mg/kg b.w.) via the jugular vein. Treatment with gID was initiated 5 days after the administration of MNU. Rats were divided into two groups (gID and non-gID), with 9 animals in the gID group and 11 animals in the non-gID group. Each rat was palpated for mammary tumors twice each week and weighed once every week. At day 170, the rats were euthanized prior to which 24-h urines were collected over dry-ice. Mammary tumors were excised, weighed and processed for histological classification.

### Metabolomics Study

Urines were deproteinized by the addition of four volumes of ice-cold aqueous methanol. Precipitated protein was removed by centrifugation at 10,000× *g* for 5 min at 4 °C; the supernatants were aspirated and evaporated to dryness under N_2_. Dried extracts were re-suspended in 0.1% formic acid (100 μL) and aliquots (10 μL) analyzed by nanoLC-MS/MS on an Eksigent reverse-phase C_18_ ChipLC column (15 cm × 200 µm i.d.) using a 20 min, 0–95% linear gradient of acetonitrile in 0.1% formic acid at 45 °C. Eluates were passed into the nano-electrospray ionization source of a SCIEX 5600 TripleTOF mass spectrometer. Data were recorded in the negative and positive ion modes. Each duty cycle consisted of a 250 ms hi-res TOF-MS scan (50–1000 *m*/*z*), followed by twenty 50 ms MSMS scans on selected precursor ions using a 15–35 V accelerating voltage. Column flow rate was 1000 nL/min, desolvation temperature was set to 120 °C and IonSpray™ and curtain gas were at 10 psi and 25 psi, respectively.

Ion features in collected data were aligned and integrated using MS-DIAL (version 4.80) and MSMS data mapped against public negative ion and positive ion MS and MSMS databases (version 15). Only compounds with MSMS spectra matching the library MSMS spectra with scores of 80 or more were used for further statistical analysis. Downloaded data files were processed to remove metabolites that either did not bind to the reverse-phase column or appeared during the period where the column was washed with 100% acetonitrile. In addition, peaks with areas less than 100 were removed before univariate and multivariate statistical analysis with MetaboAnalyst 5 (https://www.metaboanalyst.ca, accessed on 18 September 2022). Data were first normalized by the total ion current of annotated ion features and then mean-centered and subjected to Pareto scaling. Unsupervised principal component analysis (PCA) and supervised partial least squares-discriminant analysis (PLS-DA) were performed; heat maps were generated to visualize hierarchical clustering. While both PCA and PLS-DA reduce the dimension of the data, the former was used for visualizing patterns, the latter to remove uncontrolled variables and further differentiate between gID and non-gID groups [18,19]. Variable importance in the projection (VIP) scores from PLS-DA were used as estimate of the contribution of each variable to the group separation. Normalized data were also analyzed by Mummichog (version 1.0.9) to identify differentially expressed metabolic pathways.

## 3. Results

### 3.1. Cancer Chemoprevention

A preliminary experiment resulting from the change from the non-gID to gID diets revealed a 36% fall in MNU-induced mammary tumors in female Sprague Dawley rats. Since the two diets were not from the same batch, the study was repeated using a single batch of NIH-7001 diet that was divided into two–one that was exposed to γ-irradiation (20–50 kGy) and the other which was not exposed to γ-irradiation. Compared to non-gID, gID suppressed the number of palpable tumors per animal to 4.5 compared to 7 for non-gID diet group, indicating that γ-irradiation of the NIH 7001 diet caused a 36% reduction in the tumor number. This finding confirmed the preliminary observation and, although not statistically proven, suggests that irradiation of the NIH 7001 diet may have a significant role in chemoprevention in this model.

### 3.2. Multivariate Statistical Analysis

In order to capture global metabolic changes, unsupervised PCA and supervised PLS-DA multivariate statistical analyses were performed on mass spectrometry data obtained from urine samples from both groups of rats. Data normalization is critically important for accurate identification and quantification of metabolites present in various samples. We used the total ion count (TIC) method in which all metabolites in a sample are divided by the TIC observed in the sample [20,21]. These analyses showed the separation of two groups (gID and non-gID groups) based on the 95% confidence limits for both the positive and negative ions (Figure 1 and Appendix A). VIP scores were used to determine the metabolites most contributing to the group separations. There were 25 metabolites with VIP scores greater than 1 across the groups (Figure 2). A heat map of metabolites (top 25) demonstrated the relative abundances of ions in each sample and indicated that changes largely consist of increases in metabolite levels in the gID group (Appendix A).

### 3.3. Pathway Analysis

Mummichog analysis of positive ions revealed significant involvement of pathways of fatty acids, amino acids (cysteine, histidine, lysine, and methionine), and purine metabolism. Analysis of negative ions also revealed significant effects on fatty acid metabolism (in particular, butanoate and linoleate), as well as metabolism of amino acids (isoleucine and leucine, lysine, valine, and tryptophan (Appendix A). Thus, chronic intake of ID led to metabolite alteration of the urinary metabolic profile and disturbances in several other protein and energy pathways.

### 3.4. Differentially Expressed Metabolites

Because identification of differentially expressed ion features is critically important to draw any biological meaning from the analyzed data, we putatively annotated ion features with VIP scores >1 to name them as metabolites or characterized their compound class by manually matching the accurate masses of precursor ions, and their MS/MS product ion spectra to references reported in scientific publications and/or public metabolite databases. Lists of annotated metabolites are presented in Figure 2 and Figure 3. Consumption of gID increased the levels of the gut microbial metabolites phenylacetylglycine, indoxyl sulfate, kynurenic acid, hippurate and betaine. In addition, chronic intake of gID also increased the levels of riboflavin (vitamin B2), pantothenic acid (vitamin B5) and dietary components such as daidzein glucuronide.

### 3.5. Metabolite Annotation

In untargeted metabolomic studies, metabolites are annotated with 4 different levels (1–4) of confidence [22]. Level 1 includes identified and verified metabolites. Other levels are 2 (putatively identified compounds), 3 (putatively characterized compound class) and 4 (unidentified). In this study, identification levels are 2 and 4. Statistical analysis was only performed on annotated metabolite ions from level 2. Next we discuss identification of major metabolites whose levels are altered after chronic exposure to gID using MS-DIAL as well as manual interpretation of their exact masses and product ions [23].
Glycine conjugates*m/z* 194.079 [M + H]^+^, 192.065 [M-H]^−^, Rt 13.5 minMS/MS of *m/z* 194.079 contained intense fragment ions *m/z* 119.050 and 91.054 due to the neutral losses of 75.029 (NH_2_CH_2_COOH) and 103.025 Da (CONH_2_CH_2_COOH), respectively (Appendix A). In addition, the characteristic product ion *m/z* 76.040 further indicated the presence of the glycine moiety. Based on the accurate mass and product ion pattern analyses, this metabolite was annotated as phenylacetylglycine. It is the most significantly altered and upregulated urinary metabolite among the top 25 identified by PLS-DA following chronic exposure to gID in animals.*m/z* 180.066 [M + H]+, Rt 12.6 minMS/MS spectrum of the precursor ion *m/z* 180.066 contained a product ion *m/z* 162.055 due to the neutral loss of H_2_O. The observation of an intense product ion *m/z* 105.031 with a neutral 75.035 Da loss from the precursor ion indicated the presence of glycine residue [24]. Based on these data, this metabolite ion was annotated as hippurate.*m/z* 170.041 [M + H]+, Rt 9.7 minUpon MS/MS fragmentation, the protonated precursor ion *m/z* 170.041 produced several product ions characteristic of N-acylglycine. A neutral loss of 75.029 Da from the precursor ion generated an intense product ion *m/z* 95.012. In addition, it also showed product ions *m/z* 152.034 and 124.039 due the losses of 18 Da (H_2_O) and 46 Da (HCOOH), respectively. Based on these data, this metabolite ion was annotated as N(2-furoyl)glycine. Glycine N-acyltransferase catalyzes the transfer of acyl group from the acyl-CoA to glycine in the mitochondria [25]. Although glycine can be endogenously formed, composition of an animal diet (irradiated vs. non-irradiated) may influence their levels.Glycine betaine*m/z* 118.083, Rt 4.3 minThe precursor ion *m/z* 118.083 in the positive ion mode generated major product ions, *m/z* 58.069 and 59.076, with almost equal intensities (Appendix A). These are characteristic product ions of betaine and correspond to C_3_H_8_N^+^ and (CH_3_)_3_N^+^ ions, respectively [26]. This metabolite was elevated in the gID group.Glycocholate*m/z* 464.299 [M−H]^−^, Rt 20.0 minMS/MS fragmentation of *m/z* 464.299 generated the product ions *m/z* 402.297 and 74.027, after neutral losses of 62.00 Da (H_2_O + CO_2_) and glycine moiety, respectively. Comparing these data with those of a published report [27], this metabolite was annotated as a glycine conjugate of cholic acid.Riboflavin (vitamin B2)*m/z* 377.146 [M + H]+, Rt 11.9 minMS/MS fragmentation of *m/z* 377.146 generated the product ions *m/z* 359.138, 341.124 and 243.087, characteristic of riboflavin (Appendix A) [28]. Riboflavin was elevated in the urines of the gID group compared to nonID group.Pantothenic acid (vitamin B5)*m/z* 220.117 [M + H]+, Rt 9.25 minIts product ion *m/z* 184.095 occurs after the sequential losses of two H_2_O molecules. Product ions *m/z* 142.084, 124.074, 90.055 and 72.046 were also observed in the MS/MS spectrum (Appendix A). The observed characteristic product ions *m/z* 90.055 and 72.046 corresponded to the β-alanine residue [29]. Based on this evidence, the structure of this metabolite is annotated as pantothenic acid. Bacteria in the gut can produce this vitamin and its increased levels in gID fed animals may be associated with breast cancer prevention [30].Glucuronide conjugates*m/z* 431.094 [M + H]^+^, Rt 12.7 minMS/MS spectrum of the precursor ion *m/z* 431.094 contained an intense product ion *m/z* 255.065 due to a neutral loss of 176.025 Da, indicating the presence of a glucuronic acid moiety [31]. In addition, the presence of product ions *m/z* 177.050 and 149.074 enabled annotation of this metabolite ion as daidzein glucuronide (cal. *m/z* 431.097). The NIH 7001 diet contains soy protein and therefore contributed expected glucuronide metabolites of the isoflavones, daidzein and genistein.Creatine*m/z* 132.074 [M + H]+, Rt 4.4 minIn an MS/MS experiment, the precursor ion *m/z* 132.074 generated prominent product ions *m/z* 114.064 and 90.055 due to neutral losses of H_2_O and NHCNH (42.02 Da), respectively (Appendix A). Comparison of these product ion features with previously published data [32] enabled the annotation of this metabolite ion as creatine (cal. *m/z* 132.077).Creatinine*m/z* 114.065 [M + H]^+^, Rt 4.2 minThis precursor ion *m/z* 114.065 had the product ion *m/z* 86.073 after the neutral loss of CO (27.995 Da). In addition, the product ion *m/z* 72.046 is characteristic of metabolites with a guanidino group [32]. These data allowed annotation of this metabolite as creatinine (cal. *m/z* 114.066).Kynurenic acid*m/z* 190.047 [M + H]^+^, Rt 11.2 minThe precursor ion 190.047 generated the product ions *m/z* 172.027 and 162.055 following the neutral losses of H_2_O and CO, respectively. The intense product ion *m/z* 144.042 was due to loss of 46.005 Da (H_2_O + CO_2_). Other product ions included *m/z* 116.051, 89.040, and 70.070 (Appendix A). These characteric features in the MS/MS spectrum of *m/z* 190.047 allowed annotation of this metabolite ion as kynurenic acid. In this study, gID-fed animals had increased levels of this metabolite in their urine.Indoxyl-sulfate*m/z* 212.003 [M−H]^−^, 13.2 minIts precursor ion generated the major product ions *m/z* 132.046, 112.040, and 79.958. The product ion *m/z* 79.958 corresponds to a sulfate group attached to aromatic ring in the molecule. Indoxyl-sulfate is a gut microbial metabolite of tryptophan produced from sulfonation of indole by hepatic sulfotransferase [33].Citramalate*m/z* 147.029 [M−H]^−^, Rt 7.3 minUpon dissociation, it had a series of product ions *m/z* 129.019, 101.026, 103.054, and 85.030 due to neutral losses of H_2_O, HCOOH, CO_2_, and H_2_O + CO_2_, respectively. Based on accurate mass measurement, and comparison of experimental MS/MS fragmentation spectrum with the published report [34], the peak Rt 7.3 min with *m/z* 147.029 is annotated as citramalate (Appendix A). The level of citramalate is higher in samples derived from animals exposed to gID, compared to non-irradiated diet fed animals.TCA cycle intermediatesMS/MS of the precursor ion *m/z* 117.018 ([M−H]^−^, Rt 7.2 min) contained the prominent product ion *m/z* 73.031 along with *m/z* 99.010. The product ion *m/z* 73.031 is considered as a characteristic product ion of succinate.The precursor ion *m/z* 133.013 ([M−H]^−^, Rt 4.6 min) produced the major product ions *m/z* 115.003, 89.024, 71.016, and 59.019. The product ions *m/z* 115.003 and 89.024 are due to neutral losses of 18.010 (H_2_O) and 43.990 (CO_2_), respectively. These fragmention patterns correspond to malate.Similarly, MS/MS of the precursor ion *m/z* 191.021 ([M−H]^−^, 6.3 min) contained the product ions *m/z* 173.008, 147.030, 129.019, 111.009, 87.010, 67.021. Based on these data, this metabolite was annotated as citrate (Figure 3).

## 4. Discussion

In this study we report the effects of long-term consumption of γ-irradiated NIH 7001 diet in female Sprague Dawley rats treated with the carcinogen MNU. A single dose of MNU induces mammary tumors in these rats and is considered a useful animal model of mammary cancer in which to study chemoprevention [35,36]. As observed in a previous study when the use of γ-irradiated NIH 7001 diet began, γ-irradiated NIH 7001 diet also inhibited the development of mammary tumors compared to non-irradiated NIH 7001. Although the inhibition was similar (36% and 38%) in each study, this alone cannot be considered significant and further, much larger studies to verify this observation are warranted.

Because results from our previous study [13] indicated that γ-irradiation increases oxidation of unsaturated fatty acids in the NIH 7001 diet, it was important to determine how it was reflected in the metabolites excreted in the urines of treated animals. Previous studies have shown that several metabolic altered pathways are involved in mammary carcinogenesis [37,38]. Urinary metabolites, being end products of cellular regulatory processes represent a molecular phenotype. Changes in metabolite concentration are indicative of biochemical alterations induced by mammary cancer. The present study reveals that there are significant changes in urine metabolic profiles and disturbances in amino acid and energy pathways in rats fed the γ-irradiated NIH 7001 diet (Appendix A).

In this study, we used a nano-LCMS platform to analyze the urinary metabolome. Nano-LCMS/MS has widely been utilized by many investigators for untargeted metabolomics as it provides needed sensitivity to detect low abundant metabolite in a biological sample [39]. We observed increased levels of acylglycines (phenylacetylglycine and hippurate) in animals fed with gID. The variation in the availability of free glycine and CoA could potentially influence cancer development and mitochondrial energy metabolism [40,41]. Higher levels of N-acylglycine in the gID fed urine samples may also reflect increased activity of glycine N-acyltransferase. Reduced levels of glycine N-acyltransferase have been reported in human breast cancer metastasis and progression [42,43]. Animals fed with gID diet showed higher levels of glycine betain, compared to non-gID group. High dietary intake of choline and betaine showed a reduced risk of breast cancer mortality in a population based study [44].

Vitamin B2, being an essential micronutrient, is known to be involved in energy metabolism, DNA and other critical functions [45]. We observed increased levels of vitamin B2 in animals with gID. Results from previous meta-analysis have indicated that intake of riboflavin is weakly associated with reduced risk of breast cancer in humans [46]. Phase II metabolism such as glucuronidation is the main route of metabolism of many endogenously formed or ingested food components. Glucuronidation may affect bioavailability of chemopreventive agents, since it makes them more hydrophilic and eliminates them in urine. In our studies, levels of daidzein glucuronide increased significantly in the rat urine samples fed with gID. Interestingly, soy isoflavones such as daidzein exist in the circulation and in human breast tissues predominantly as their glucuronides [47].

In the present studies, gID-fed animals also showed increased levels of kynurenic acid in their urine samples. Previous studies have shown that kynurenic acid induces cell cycle arrest and influences tumor growth [48].

Examination of the fecal microbiome from rats on the irradiated diet revealed substantial changes in the pattern of microorganisms compared to the non-irradiated diet (unpublished results). Alteration of specific metabolites or microbiota may either predispose to mammary cancer or be acquired during the process of cancer progression. Although it is unclear which metabolites are absorbed and bioavailable in the mammary tissues and elicit anti-carcinogenic effect, the additive and or synergistic effects of altered metabolites may be responsible for their cancer chemopreventive effects in this study.

Our central hypothesis is that chronic exposure to an irradiated diet to animals alters the composition and metabolic activity of the gut microbiota, and or reverses metabolic reprograming of energy metabolism of mammary cancer cells which in turn impact mammary cancer susceptibility and progression. These preliminary studies also indicated changes in the gut microbiota and urinary metabolome in rats exposed to irradiated diet. Because about 20% of altered metabolites in VIP score plot analysis are derived from or modified by the gut microbiota in the present study, these metabolites may exert their effects as signaling molecules and substrate of metabolic reactions [49]. Specific metabolites are implicated in the development and prevention of cancers. Medicinal properties of food depend on in part, the gut microbiota composition and metabolism of endogenous and exogenous compounds. It is becoming clear that alteration of specific microbiota may either predispose to mammary cancer or be acquired during the process of cancer progression or prevention [50,51]. Examination of the fecal microbiota from rats on the irradiated diet revealed substantial changes in the pattern of the gut microbiota compared to the non-irradiated diet (unpublished results).

## 5. Conclusions and Future Directions

Overall, our data indicate that consumption of the γ-irradiated diet led to unexpectedly a 36% reduction in induced-mammary tumors in rats and alteration of a number of metabolites, including the gut microbial metabolites such as phenylacetylglycine, indoxyl sulfate, kynurenic acid, hippurate and betaine in the urine. These metabolites have the potential to be biomarkers associated with breast cancer and help understand the tumor microenvironment and pathogenesis. It is important to investigate what metabolites are bioavailable in mammary tissues using mass spectrometry because the metabolic nature of mammary tissue may indicate the cancer progression or prevention. Because γ-irradiation results in compositional changes of food components, [13], how variable doses of irradiation (40–50 and 10–20 kGy) during exposure affect the levels of food components and chemoprevention warrants further investigations. In addition, future lipidomics studies should be designed to examine tumor lipid phenotype in response to chronic intake of irradiated diet in the rat model of N-methyl-N-nitrosourea (MNU)-induced mammary tumors.

## Figures and Tables

**Figure 1 metabolites-12-00976-f001:**
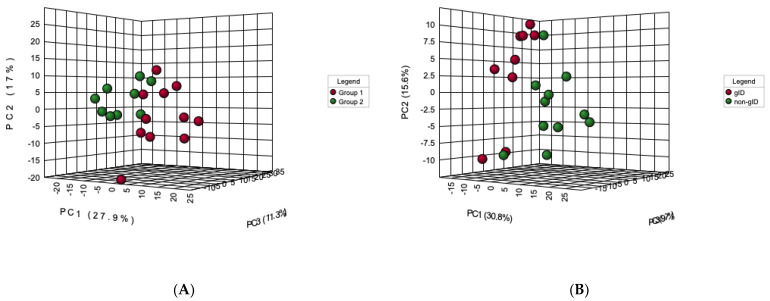
3D−PCA score plots of urine samples from gID fed animals (red spheres) and non−gID fed animals (green spheres). This analysis shows data from negative (**A**) and positive (**B**) ion modes. This separation was also clearly observed in supervised PLS−DA plots (Appendix A).

**Figure 2 metabolites-12-00976-f002:**
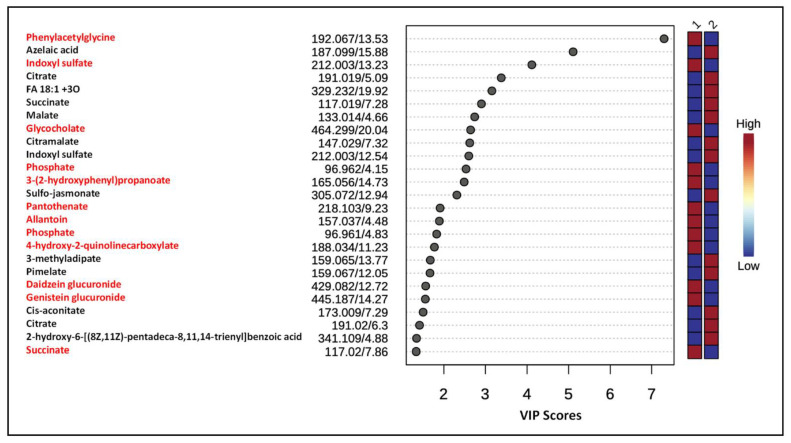
Variable Importance of Projection (VIP) scores for the 25 features most contributing to the PLS-DA plots in the negative ion mode. Colored boxes on the right side indicate the relative intensity of the corresponding feature in each group. ND = not determined. 1 = gID−fed animals, 2 = non−gID-fed animals. Metabolites highlighted in red indicate their higher levels in animals fed with gID.

**Figure 3 metabolites-12-00976-f003:**
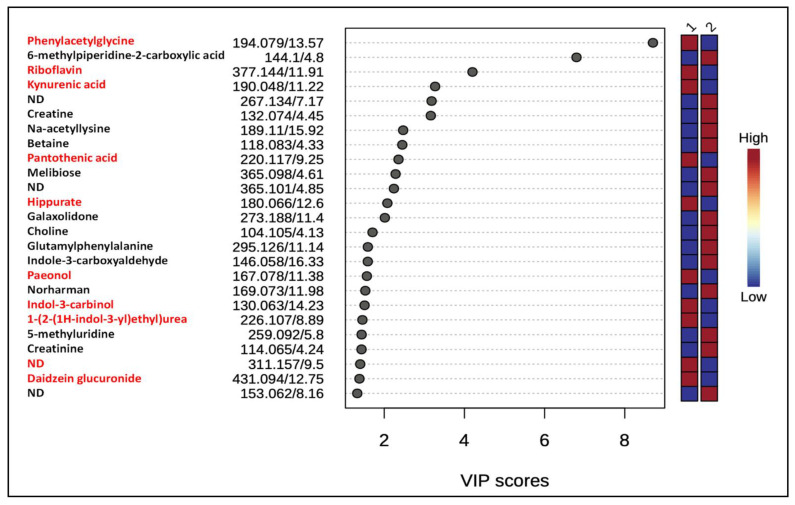
Variable Importance of Projection (VIP) scores for the 25 most contributing features from PLS−DA in positive ion modes. Colored boxes on the right panel indicate the relative intensity of the corresponding feature in each group. ND = not determined. 1 = gID−fed animals, 2 = non−gID-fed animals. Metabolites highlighted in red indicate their higher levels in animals fed with gID.

## Data Availability

The data presented in this study are available in the main article and the Appendix A.

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
