# Peer review of "Gamma-ray Irradiation of Rodent Diets Alters the Urinary Metabolome in Rats with Chemically Induced Mammary Cancer"

_metabolites, 2022, doi:10.3390/metabo12100976_

Round 1
Reviewer 1 Report
Thank you very much for inviting us to review the article entitle “Gamma-ray irradiation of rodent diets alters the urinary metabolome in rats with chemically induced mammary cancer” (metabolites-1951949). This manuscript is presented to evaluate its suitability publication in Metabolites in the Special Issue “Cancer Metabolics 2023”.
This study presented comparative untargeted metabolomics approach was applied to compare urinary metabolite profiles of rats fed irradiated and non-irradiated diets. Its try to understand whether consumption of gID affects the urinary metabolite profile, which by implication may be associated with increased cancer risk or prevention.
This study concludes that metabolic changes in rat urine in response to an irradiated diet may be associated with the prevention of breast cancer.
The summary is clear and well structured and provides relevant information about the content of the work.
Introduction is based on previous relevant studies to lay the foundations of the research carried out. The bibliography is adequate. I suggest that the objective be rewritten more clearly at the end of the introduction.
The experimental design must have the approval of the ethical committee for animal experimentation. Finally, 9 animals participated in the irradiated group and 11 animals in the non-irradiated group.
The results are presented with the discussion. Really, I suggest that results and discussion be separated for a better understanding of the contribution made by the authors. Practically all of this section is about results and there is no discussion in which comparisons with other studies or limitations and strengths of the study itself are proposed.
Future research derived from the results of this study should be under discussion, leaving the conclusions section as the contribution of the study and not the methodology used.
Reviewer 2 Report
Prasain et al have performed what I find to be a very interesting study looking at the role of gamma irradiated food and the effects on mammary cancer in rats. The work done is very good and the authors have done an amazing job of describing the putative metabolites they ID and show the MS spectra which is something very few people do so I am very grateful for this and it gives a lot of credence to their work. This work will be of great interest to the readers of metabolites. Though I feel that there needs to be more back ground information in the introduction to set the scene more. At the moment it is aimed at specialists in chemopreventive diets and needs to appeal to a wider metabolomics audience who don’t have this disease specific background. There is also some nice metabolomics technology being used which is vastly underused in the field and I feel warrants further discussion. However, I believe the authors will be able to make these changes relatively easily and the resulting manuscript would have a wider appeal.
Abstract
The gamma sign is mis-printed and requires alteration
Introduction
The introduction is too brief it is not clear why this work is being done. Can the authors please explain the following:
1) Why is there interest in gamma irradiated food for cancer prevention?
2) Why mammary cancer?
3) What is metabolomics and why is it used
a. Why use nanoLC-MS? I am a huge proponent of this, but it is so rarely used and it would be really useful for the authors to describe why they used nano scale LC-MS, the advantages of it vs conventional or microflow and cite some of the relevant literature on nanoLC-MS for metabolomics and specifically urine metabolomics to show the validity of this approach and that it is reproducible.
3. Metabolomics study
Urinary concentration varies quite considerably and as such requires normalisation. I notice that only total ion signal normalisation was used here. Why was this? Do the authors think this may introduce some bias?
Given this is nanoLC can the authors add details such as the flow rate, desolvation temperature and gas flow e.t.c.
It is probably appropriate to also cite the relevant MetaboAnalyst and MSDIAL papers
Results and discussion
General comment: How do the authors differentiate between metabolites caused by irradiated diet vs degree of tumours detected? Could the variation seen be due to the different numbers of tumours seen? I think some of this could also be clarified with the suggested changes to the introduction.
Figure 1: This is missing error bars
Figure 2: If the PCA shows the same separation it would be much better to show this as PCA is unsupervised and thus less biased compared to a PLS-DA. I also notice there are no QCs shown on these scores plots, were QC samples run? If not how have the authors assessed the reproducibility of the analysis? If they have can they include them on the PCA scores plot to show they group tightly.
Figure 3 and 4: What is the significance of the metabolites highlighted in red? This needs to be detailed in the legends.
Sections 4.6-4.16: I appreciate that the authors are stating that these are putative IDs and have detailed why they believe them to be the ID they have assigned. Showing the MS spectra in the supplemental is also a really really nice feature that more people should adopt. This is a really nice change from traditional metabolomics papers and I am big fans of the authors for making this distinction and presenting all of their evidence. However, I think it would be nice to discuss the potentially significance of these metabolites too rather than just describing how they were annotated. Why would citrate be significant for example or indole-sulfate e.t.c. There is room here for a more extended and interesting discussion.
Will all of the data be uploaded to a metabolomics database such as metabolights?
Round 2
Reviewer 1 Report
After reviewing the new version of the manuscript entitled "Gamma-ray irradiation of rodent diets alters the urinary metabolome in rats with chemically induced mammary cancer" (metabolites-1951949), as well as the comments of the authors to the suggestions made, I been able to verify that the changes made by the authors, following both the recommendations that I have made and those made by the other reviewer, have substantially improved the quality of the article.
Minor comment:
In section two on methodology, it would be convenient to include the approval number of the ethics committee.
The methodology section in relation to the analysis of the information should be expanded in order to better understand the results.
Reviewer 2 Report
The authors have responded to my queries well and I believe present an improved paper that will be of interest to readers of metabolites.
The only query I have remaining is that its stated that histology of the tumours was done but there are no results for this presented here. This would be an interesting thing to have being as the data is presumably available. If the tumours are worse for the gID foodstuff then that would be very useful to know for example. So maybe a table detailing the histological findings or even an image or two of them. This would all fit in the supplementary if the authors didn’t want it in the main manuscript but I feel it’s important to add.
